# Are Compressed Language Models Less Subgroup Robust?

**Leonidas Gee**[1*]     **Andrea Zugarini**[4]     **Novi Quadrianto**[1,2,3]

[1]Predictive Analytics Lab, University of Sussex, UK
[2]BCAM Severo Ochoa Strategic Lab on Trustworthy Machine Learning, Spain
[3]Monash University, Indonesia
[4]expert.ai, Siena, Italy

## Abstract

To reduce the inference cost of large language models, model compression is increasingly used to create smaller scalable models. However, little is known about their robustness to minority subgroups defined by the labels and attributes of a dataset. In this paper, we investigate the effects of 18 different compression methods and settings on the subgroup robustness of BERT language models. We show that worst-group performance does not depend on model size alone, but also on the compression method used. Additionally, we find that model compression does not always worsen the performance on minority subgroups. Altogether, our analysis serves to further research into the subgroup robustness of model compression.

## 1 Introduction

In recent years, the field of Natural Language Processing (NLP) has seen a surge in interest in the application of Large Language Models (LLMs) (Brown et al., 2020; Thoppilan et al., 2022; Touvron et al., 2023). These applications range from simple document classification to complex conversational chatbots. However, the uptake of LLMs has not been evenly distributed across society. Due to their large inference cost, only a few well-funded companies may afford to run LLMs at scale. To address this, many have turned to model compression to create smaller language models (LMs) with near comparable performance to their larger counterparts.

The goal of model compression is to reduce a model's size and latency while retaining overall performance. Existing approaches such as knowledge distillation (Hinton et al., 2015) have produced scalable task-agnostic models (Turc et al., 2019; Sanh et al., 2020; Jiao et al., 2020). Meanwhile, other approaches have shown that not all transformer

---

* Corresponding author: jg717@sussex.ac.uk.

| Model | Size (MB) | Parameters |
|---|---|---|
| BERT$_{Base}$ | 438.01 | 109M |
| BERT$_{Medium}$ | 165.55 | 41M |
| BERT$_{Small}$ | 115.09 | 29M |
| BERT$_{Mini}$ | 44.71 | 11M |
| BERT$_{Tiny}$ | 17.56 | 4M |
| DistilBERT | 267.86 | 67M |
| TinyBERT$_6$ | 267.87 | 67M |
| TinyBERT$_4$ | 57.43 | 14M |
| BERT$_{PR20}$ | 415.97 | 104M |
| BERT$_{PR40}$ | 393.14 | 98M |
| BERT$_{PR60}$ | 370.31 | 93M |
| BERT$_{PR80}$ | 347.48 | 87M |
| BERT$_{DQ}$ | 181.48 | 24M |
| BERT$_{SQ}$ | 182.89 | 24M |
| BERT$_{QAT}$ | 182.89 | 24M |
| BERT$_{VT100}$ | 438.01 | 109M |
| BERT$_{VT75}$ | 414.57 | 104M |
| BERT$_{VT50}$ | 391.13 | 98M |
| BERT$_{VT25}$ | 367.69 | 92M |

Table 1: Model size and number of parameters. BERT$_{Base}$ is shown as the baseline with subsequent models from knowledge distillation, structured pruning, quantization, and vocabulary transfer respectively.

heads (Michel et al., 2019) or embeddings (Gee et al., 2022) are essential. Although model compression has been proven to work well in practice, little is known about its influence on subgroup robustness.

In any given dataset, subgroups exists as a combination of labels (e.g. hired or not hired) and attributes (e.g. male or female) (Sagawa et al., 2020; Bartlett et al., 2022). A model is said to be subgroup robust if it maximizes the lowest performance across subgroups (Gardner et al., 2023). Due to the unbalanced sample size of each subgroup, the conventional approach to training via Empirical Risk Minimization (ERM) (Vapnik, 1999) produces models with a higher performance

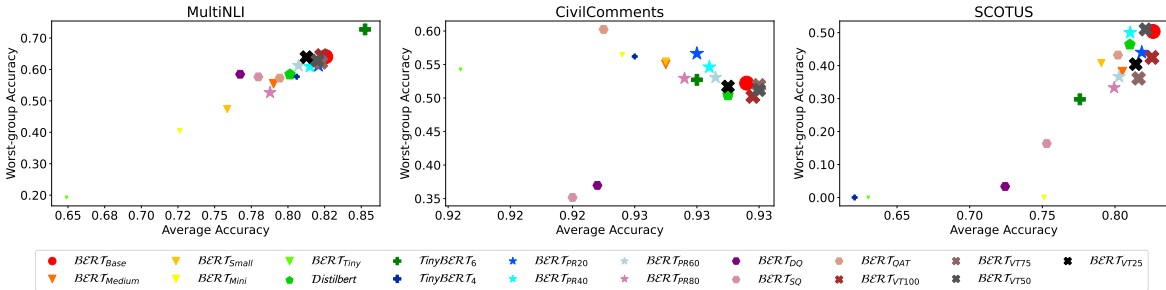

Figure 1: Plot of WGA against average accuracy. Compression method is represented by marker type, while model size is represented by marker size. In MultiNLI and SCOTUS, compression worsens WGA for most models. Conversely, WGA improves for most compressed models in CivilComments.

on majority subgroups (e.g. hired male), but a lower performance on minority subgroups (e.g. hired female).

Given the increasing role of LLMs in everyday life, our work seeks to address a gap in the existing literature regarding the subgroup robustness of model compression in NLP. To that end, we explore a wide range of compression methods (Knowledge Distillation, Pruning, Quantization, and Vocabulary Transfer) and settings on 3 textual datasets — MultiNLI (Williams et al., 2018), CivilComments (Koh et al., 2021), and SCOTUS (Chalkidis et al., 2022). The code for our paper is publicly available[1].

The remaining paper is organized as follows. First, we review related works in Section 2. Then, we describe the experiments and results in Sections 3 and 4 respectively. Finally, we draw our conclusions in Section 5.

## 2 Related Works

Most compression methods belong to one of the following categories: Knowledge Distillation (Hinton et al., 2015), Pruning (Han et al., 2015), or Quantization (Jacob et al., 2017). Additionally, there exists orthogonal approaches specific to LMs such as Vocabulary Transfer (Gee et al., 2022). Previous works looking at the effects of model compression have focused on the classes or attributes in images.

Hooker et al. (2021) analyzed the performance of compressed models on the imbalanced classes of CIFAR-10, ImageNet, and CelebA. Magnitude pruning and post-training quantization were considered with varying levels of sparsity and precision respectively. Model compression is found to cannibalize the performance on a small subset of classes

to maintain overall performance.

Hooker et al. (2020) followed up by analyzing how model compression affects the performance on sensitive attributes of CelebA. Unitary attributes of gender and age as well as their intersections (e.g. Young Male) were considered. The authors found that overall performance was preserved by sacrificing the performance on low-frequency attributes.

Stoychev and Gunes (2022) expanded the previous analysis on attributes to the fairness of facial expression recognition. The authors found that compression does not always impact fairness in terms of gender, race, or age negatively. The impact of compression was also shown to be non-uniform across the different compression methods considered.

To the best of our knowledge, we are the first to investigate the effects of model compression on subgroups in a NLP setting. Additionally, our analysis encompasses a much wider range of compression methods than were considered in the aforementioned works.

## 3 Experiments

The goal of learning is to find a function $f$ that maps inputs $x \in X$ to labels $y \in Y$. Additionally, there exists attributes $a \in A$ that are only provided as annotations for evaluating the worst-group performance at test time. The subgroups can then be defined as $g \in \{Y \times A\}$.

### 3.1 Models

We utilize 18 different compression methods and settings on BERT (Devlin et al., 2019). An overview of each model's size and parameters is shown in Table 1.

**Knowledge Distillation (KD).** We analyze seven models (BERT$_{Medium}$, BERT$_{Small}$,

---

[1] https://github.com/wearepal/compression-subgroup

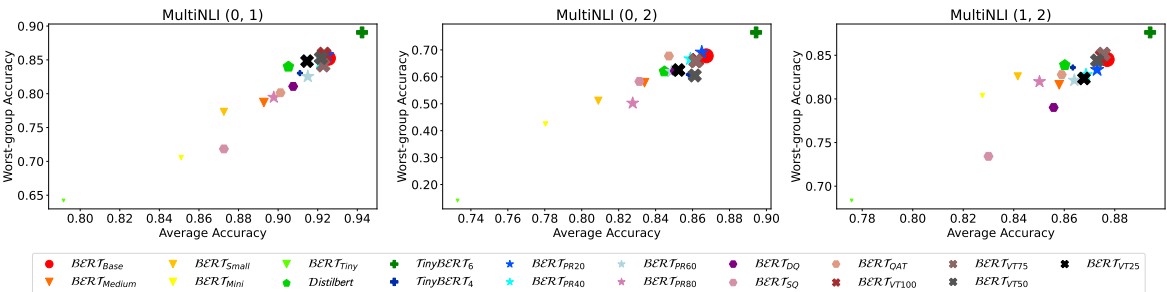

Figure 2: Model performance is shown to improve across the binary datasets of MultiNLI. However, the overall trend in WGA remains relatively unchanged, with a decreasing model size leading to drops in WGA.

$BERT_{Mini}$, $BERT_{Tiny}$, DistilBERT, TinyBERT$_6$, TinyBERT$_4$) distilled from the uncased version of $BERT_{Base}$ using 3 different distillation methods (Turc et al., 2019; Sanh et al., 2020; Jiao et al., 2020). Each model is loaded from HuggingFace[2] with its pre-trained weights.

**Pruning.** We analyze structured pruning (Michel et al., 2019) on BERT following a three-step training pipeline (Han et al., 2015). Four different levels of sparsity ($BERT_{PR20}$, $BERT_{PR40}$, $BERT_{PR60}$, $BERT_{PR80}$) are applied by sorting all transformer heads using the L1-norm of weights from the query, key, and value projection matrices. Structured pruning is implemented using the NNI library[3].

**Quantization.** We analyze 3 quantization methods supported natively by PyTorch — Dynamic Quantization ($BERT_{DQ}$), Static Quantization ($BERT_{SQ}$), and Quantization-aware Training ($BERT_{QAT}$). Quantization is applied to the linear layers of BERT to map representations from FP32 to INT8. The calibration required for $BERT_{SQ}$ and $BERT_{QAT}$ is done using the training set.

**Vocabulary Transfer (VT).** We analyze vocabulary transfer using 4 different vocabulary sizes ($BERT_{VT100}$, $BERT_{VT75}$, $BERT_{VT50}$, $BERT_{VT25}$) as done by Gee et al. (2022). Note that $BERT_{VT100}$ does not compress the LM, but adapts its vocabulary fully to the in-domain dataset, thus making tokenization more efficient.

### 3.2 Datasets

Our analysis is done on 3 classification datasets. MultiNLI and CivilComments are textual datasets used by most subgroup robustness research (Sagawa et al., 2020; Liu et al., 2021; Izmailov et al., 2022). Additionally, we extend

the datasets to SCOTUS from the FairLex benchmark (Chalkidis et al., 2022).

Further details regarding the subgroups in each dataset are shown in Appendix A.1.

**MultiNLI.** Given a hypothesis and premise, the task is to predict whether the hypothesis is contradicted by, entailed by, or neutral with the premise (Williams et al., 2018). Following Sagawa et al. (2020), the attribute indicates whether any negation words (*nobody*, *no*, *never*, or *nothing*) appear in the hypothesis. We use the same dataset splits as Liu et al. (2021).

**CivilComments.** Given an online comment, the task is to predict whether it is neutral or toxic (Koh et al., 2021). Following Koh et al. (2021), the attribute indicates whether any demographic identities (*male*, *female*, *LGBTQ*, *Christian*, *Muslim*, *other religion*, *Black*, or *White*) appear in the comment. We use the same dataset splits as Liu et al. (2021).

**SCOTUS.** Given a court opinion from the US Supreme Court, the task is to predict its thematic issue area (Chalkidis et al., 2022). Following Chalkidis et al. (2022), the attribute indicates the direction of the decision (*liberal* or *conservative*) as provided by the Supreme Court Database (SCDB). We use the same dataset splits as Chalkidis et al. (2022).

### 3.3 Implementation Details

We train each compressed model via ERM with 5 different random initializations. The average accuracy, worst-group accuracy (WGA), and model size are measured as metrics. The final value of each metric is the average of all 5 initializations.

Following Liu et al. (2021); Chalkidis et al. (2022), we fine-tune the models for 5 epochs on

---

[2]https://github.com/huggingface/transformers
[3]https://github.com/microsoft/nni

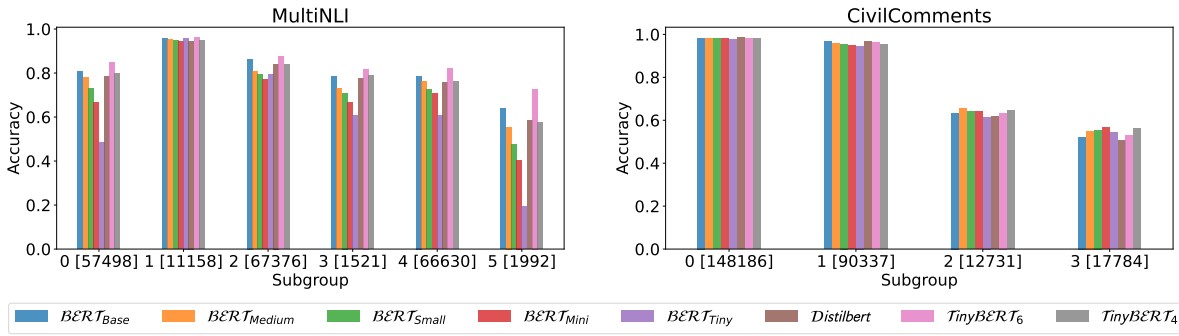

Figure 3: Distribution of accuracies by subgroup for KD. Sample sizes in the training set are shown beside each subgroup. In CivilComments, performance improves on minority subgroups (2 and 3) across most models as model size decreases contrary to the minority subgroups (3 and 5) of MultiNLI.

MultiNLI and CivilComments and for 20 on SCO-TUS. A batch size of 32 is used for MultiNLI and 16 for CivilComments and SCOTUS. Each model is implemented with an AdamW optimizer (Loshchilov and Hutter, 2019) and early stopping. A learning rate of $2 \cdot 10^{-5}$ with no weight decay is used for MultiNLI, while a learning rate of $10^{-5}$ with a weight decay of 0.01 is used for Civil-Comments and SCOTUS. Sequence lengths are set to 128, 300, and 512 for MultiNLI, CivilComments, and SCOTUS respectively.

As done by Gee et al. (2022), one epoch of masked-language modelling is applied before fine-tuning for VT. The hyperparameters are the same as those for fine-tuning except for a batch size of 8.

## 4 Results

**Model Size and Subgroup Robustness.** We plot the overall results in Figure 1 and note a few interesting findings. First, in MultiNLI and SCO-TUS, we observe a trend of decreasing average and worst-group accuracies as model size decreases. In particular, TinyBERT$_6$ appears to be an outlier in MultiNLI by outperforming every model including BERT$_{Base}$. However, this trend does not hold in CivilComments. Instead, most compressed models show an improvement in WGA despite slight drops in average accuracy. Even extremely compressed models like BERT$_{Tiny}$ are shown to achieve a higher WGA than BERT$_{Base}$. We hypothesize that this is due to CivilComments being a dataset that BERT$_{Base}$ easily overfits on. As such, a reduction in model size serves as a form of regularization for generalizing better across subgroups. Additionally, we note that a minimum model size is required for fitting the minority subgroups. Specifically, the WGA of distilled models with layers fewer

than 6 (BERT$_{Mini}$, BERT$_{Tiny}$, and TinyBERT$_4$) is shown to collapse to 0 in SCOTUS.

Second, we further analyze compressed models with similar sizes by pairing DistilBERT with TinyBERT$_6$ as well as post-training quantization (BERT$_{DQ}$ and BERT$_{SQ}$) with BERT$_{QAT}$ according to their number of parameters in Table 1. We find that although two models may have an equal number of parameters (approximation error), their difference in weight initialization after compression (estimation error) as determined by the compression method used will lead to varying performance. In particular, DistilBERT displays a lower WGA on MultiNLI and Civil-Comments, but a higher WGA on SCOTUS than TinyBERT$_6$. Additionally, post-training quantization (BERT$_{DQ}$ and BERT$_{SQ}$) which does not include an additional fine-tuning step after compression or a compression-aware training like BERT$_{QAT}$ is shown to be generally less subgroup robust. These methods do not allow for the recovery of model performance after compression or to prepare for compression by learning compression-robust weights.

**Task Complexity and Subgroup Robustness** To understand the effects of task complexity on subgroup robustness, we construct 3 additional datasets by converting MultiNLI into a binary task. From Figure 2, model performance is shown to improve across the binary datasets for most models. WGA improves the least when Y = [0, 2], i.e. when sentences contradict or are neutral with one another. Additionally, although there is an overall improvement in model performance, the trend in WGA remains relatively unchanged as seen in Figure 1. A decreasing model size is accompanied by a reduction in WGA for most models. We hypoth-

esize that subgroup robustness is less dependent on the task complexity as defined by number of subgroups that must be fitted.

**Distribution of Subgroup Performance.** We plot the accuracies distributed across subgroups in Figure 3. We limit our analysis to MultiNLI and CivilComments with KD for visual clarity. From Figure 3, we observe that model compression does not always maintain overall performance by sacrificing the minority subgroups. In MultiNLI, a decreasing model size reduces the accuracy on minority subgroups (3 and 5) with the exception of TinyBERT$_6$. Conversely, most compressed models improve in accuracy on the minority subgroups (2 and 3) in CivilComments. This shows that model compression does not necessarily cannibalize the performance on minority subgroups to maintain overall performance, but may improve performance across all subgroups instead.

## 5 Conclusion

In this work, we presented an analysis of existing compression methods on the subgroup robustness of LMs. We found that compression does not always harm the performance on minority subgroups. Instead, on datasets that a model easily overfits on, compression can aid in the learning of features that generalize better across subgroups. Lastly, compressed LMs with the same number of parameters can have varying performance due to differences in weight initialization after compression.

## Limitations

Our work is limited by its analysis on English language datasets. The analysis can be extended to other multi-lingual datasets from the recent FairLex benchmark (Chalkidis et al., 2022). Additionally, we considered each compression method in isolation and not in combination with one another.

## Acknowledgements

This research was supported by a European Research Council (ERC) Starting Grant for the project "Bayesian Models and Algorithms for Fairness and Transparency", funded under the European Union's Horizon 2020 Framework Programme (grant agreement no. 851538). Novi Quadrianto is also supported by the Basque Government through the BERC 2022-2025 program and by the Ministry of Science and Innovation: BCAM Severo Ochoa accreditation CEX2021-001142-S / MICIN/ AEI/ 10.13039/501100011033. Additionally, we would like to thank Justina Li, Qiwei Peng, and Myles Bartlett for proof reading the paper.

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

## A  Further Details

### A.1  Datasets

We tabulate the labels and attributes that define each subgroup in Table 2. Additionally, we show the sample size of each subgroup in the training, validation, and test sets.

### A.2  Results

We tabulate the main results of the paper in Table 3. The performance of each model is averaged across 5 seeds.

## B  Additional Experiments

### B.1  Sparsity and Subgroup Robustness.

Besides structured pruning, we investigate the effects of unstructured pruning using 4 similar levels of sparsity. Connections are pruned via PyTorch by sorting the weights of every layer using the L1-norm. We tabulate the results separately in Table 4 as PyTorch does not currently support sparse neural networks. Hence, no reduction in model size is seen in practice. From Table 4, we observe similar trends to those in Figure 1. Specifically, as sparsity increases, the WGA generally worsens in MultiNLI and SCOTUS, but improves in Civil-Comments across most models. At a sparsity of 80%, WGA drops significantly for MultiNLI and SCOTUS, but not for CivilComments.

### B.2  Ablation of TinyBERT$_6$.

To better understand the particular subgroup robustness of TinyBERT$_6$, we conduct an ablation on its general distillation procedure. Specifically, we ablate the attention matrices, hidden states, and embeddings as sources of knowledge when distilling on the Wikipedia dataset[4]. The same hyperparameters as Jiao et al. (2020) are used except for a batch size of 256 and a gradient accumulation of 2 due to memory constraints.

From Table 5, we find that we are unable to achieve a similar WGA on MultiNLI and its binary variants as shown by the performance gap between TinyBERT$_6$ and TinyBERT$_{AHE}$. On SCO-TUS, the WGA of TinyBERT$_{AHE}$ is found to also be much higher than TinyBERT$_6$. We hypothesize that the pre-trained weights that were uploaded to HuggingFace[5] may have included a

further in-domain distillation on MultiNLI. Additionally, model performance is shown to benefit the least when knowledge from the embedding is included during distillation. This can be seen by the lower WGA of TinyBERT$_{AHE}$ compared to TinyBERT$_{AH}$ across most datasets.

---

[4] https://huggingface.co/datasets/wikipedia
[5] https://huggingface.co/huawei-noah/TinyBERT_General_6L_768D

| Dataset | Subgroup | Label | Attribute | Training | Validation | Test |
|---|---|---|---|---|---|---|
| MultiNLI | 0 | 0 (Contradicts) | 0 (No negation) | 57498 | 22814 | 34597 |
| | 1 | 0 (Contradicts) | 1 (Has negation) | 11158 | 4634 | 6655 |
| | 2 | 1 (Entails) | 0 (No negation) | 67376 | 26949 | 40496 |
| | 3 | 1 (Entails) | 1 (Has negation) | 1521 | 613 | 886 |
| | 4 | 2 (Neutral) | 0 (No negation) | 66630 | 26655 | 39930 |
| | 5 | 2 (Neutral) | 1 (Has negation) | 1992 | 797 | 1148 |
| CivilComments | 0 | 0 (Neutral) | 0 (No sensitive) | 148186 | 25159 | 72373 |
| | 1 | 0 (Neutral) | 1 (Has sensitive) | 90337 | 14966 | 46185 |
| | 2 | 1 (Toxic) | 0 (No sensitive) | 12731 | 2111 | 6063 |
| | 3 | 1 (Toxic) | 1 (Has sensitive) | 17784 | 2944 | 9161 |
| SCOTUS | 0 | 0 (Criminal Procedure) | 0 (Liberal) | 869 | 111 | 114 |
| | 1 | 0 (Criminal Procedure) | 1 (Conservative) | 701 | 82 | 101 |
| | 2 | 1 (Civil Rights) | 0 (Liberal) | 536 | 67 | 72 |
| | 3 | 1 (Civil Rights) | 1 (Conservative) | 683 | 87 | 90 |
| | 4 | 2 (First Amendment) | 0 (Liberal) | 267 | 29 | 37 |
| | 5 | 2 (First Amendment) | 1 (Conservative) | 334 | 44 | 39 |
| | 6 | 3 (Due Process) | 0 (Liberal) | 130 | 18 | 23 |
| | 7 | 3 (Due Process) | 1 (Conservative) | 162 | 18 | 24 |
| | 8 | 4 (Privacy) | 0 (Liberal) | 66 | 5 | 6 |
| | 9 | 4 (Privacy) | 1 (Conservative) | 29 | 6 | 3 |
| | 10 | 5 (Attorneys) | 0 (Liberal) | 34 | 3 | 4 |
| | 11 | 5 (Attorneys) | 1 (Conservative) | 32 | 9 | 6 |
| | 12 | 6 (Unions) | 0 (Liberal) | 149 | 10 | 20 |
| | 13 | 6 (Unions) | 1 (Conservative) | 174 | 17 | 28 |
| | 14 | 7 (Economic Activity) | 0 (Liberal) | 644 | 87 | 70 |
| | 15 | 7 (Economic Activity) | 1 (Conservative) | 925 | 105 | 112 |
| | 16 | 8 (Judicial Power) | 0 (Liberal) | 689 | 86 | 75 |
| | 17 | 8 (Judicial Power) | 1 (Conservative) | 384 | 41 | 39 |
| | 18 | 9 (Federalism) | 0 (Liberal) | 144 | 18 | 13 |
| | 19 | 9 (Federalism) | 1 (Conservative) | 189 | 31 | 22 |
| | 20 | 10 (Interstate Relations) | 0 (Liberal) | 67 | 16 | 4 |
| | 21 | 10 (Interstate Relations) | 1 (Conservative) | 209 | 24 | 29 |

Table 2: Defined subgroups in MultiNLI, CivilComments, and SCOTUS.

| Model | MultiNLI | | CivilComments | | SCOTUS | |
|---|---|---|---|---|---|---|
| | Average | Worst | Average | Worst | Average | Worst |
| $\text{BERT}_{Base}$ | 82.58 | 64.06 | 92.96 | 52.22 | 82.62 | 50.36 |
| $\text{BERT}_{Medium}$ | 79.02 | 55.46 | 92.70 | 55.02 | 80.50 | 38.18 |
| $\text{BERT}_{Small}$ | 75.86 | 47.40 | 92.70 | 55.44 | 79.06 | 40.76 |
| $\text{BERT}_{Mini}$ | 72.62 | 40.50 | 92.56 | 56.46 | 75.12 | 0.00 |
| $\text{BERT}_{Tiny}$ | 64.90 | 19.26 | 92.04 | 54.26 | 63.02 | 0.00 |
| DistilBERT | 80.14 | 58.40 | 92.90 | 50.42 | 81.02 | 46.40 |
| $\text{TinyBERT}_6$ | 85.26 | 72.74 | 92.80 | 52.72 | 77.58 | 29.78 |
| $\text{TinyBERT}_4$ | 80.60 | 57.70 | 92.60 | 56.20 | 62.08 | 0.00 |
| $\text{BERT}_{PR20}$ | 82.08 | 61.24 | 92.80 | 56.64 | 81.84 | 44.00 |
| $\text{BERT}_{PR40}$ | 81.50 | 60.92 | 92.84 | 54.62 | 81.04 | 50.00 |
| $\text{BERT}_{PR60}$ | 80.72 | 61.24 | 92.86 | 53.08 | 80.28 | 36.64 |
| $\text{BERT}_{PR80}$ | 78.78 | 52.64 | 92.76 | 52.94 | 79.94 | 33.30 |
| $\text{BERT}_{DQ}$ | 76.74 | 58.48 | 92.48 | 36.96 | 72.44 | 3.34 |
| $\text{BERT}_{SQ}$ | 77.98 | 57.64 | 92.40 | 35.16 | 75.30 | 16.36 |
| $\text{BERT}_{QAT}$ | 79.44 | 57.26 | 92.50 | 60.24 | 80.20 | 43.20 |
| $\text{BERT}_{VT100}$ | 82.30 | 64.40 | 92.98 | 50.28 | 82.54 | 42.48 |
| $\text{BERT}_{VT75}$ | 82.26 | 62.24 | 93.00 | 51.96 | 81.62 | 36.12 |
| $\text{BERT}_{VT50}$ | 82.00 | 62.80 | 93.00 | 50.86 | 82.10 | 50.98 |
| $\text{BERT}_{VT25}$ | 81.26 | 63.92 | 92.90 | 52.14 | 81.42 | 40.44 |

(a) MultiNLI, CivilComments, and SCOTUS.

| Model | Y = [0, 1] | | Y = [0, 2] | | Y = [1, 2] | |
|---|---|---|---|---|---|---|
| | Average | Worst | Average | Worst | Average | Worst |
| $\text{BERT}_{Base}$ | 92.54 | 85.24 | 86.74 | 67.80 | 87.70 | 84.52 |
| $\text{BERT}_{Medium}$ | 89.28 | 78.70 | 83.40 | 57.84 | 85.80 | 81.62 |
| $\text{BERT}_{Small}$ | 87.26 | 77.32 | 80.90 | 51.14 | 84.16 | 82.58 |
| $\text{BERT}_{Mini}$ | 85.10 | 70.54 | 78.04 | 42.50 | 82.76 | 80.38 |
| $\text{BERT}_{Tiny}$ | 79.16 | 64.22 | 73.28 | 14.08 | 77.58 | 68.36 |
| DistilBERT | 90.52 | 84.00 | 84.48 | 62.02 | 86.02 | 83.88 |
| $\text{TinyBERT}_6$ | 94.24 | 89.06 | 89.44 | 76.50 | 89.40 | 87.62 |
| $\text{TinyBERT}_4$ | 91.10 | 83.04 | 85.80 | 60.84 | 86.34 | 83.60 |
| $\text{BERT}_{PR20}$ | 92.48 | 85.72 | 86.50 | 69.12 | 87.30 | 83.34 |
| $\text{BERT}_{PR40}$ | 92.14 | 84.64 | 85.88 | 66.64 | 86.86 | 82.76 |
| $\text{BERT}_{PR60}$ | 91.50 | 82.54 | 84.86 | 62.00 | 86.40 | 82.14 |
| $\text{BERT}_{PR80}$ | 89.78 | 79.48 | 82.76 | 50.24 | 85.02 | 81.98 |
| $\text{BERT}_{DQ}$ | 90.76 | 81.08 | 85.04 | 62.18 | 85.58 | 79.02 |
| $\text{BERT}_{SQ}$ | 87.26 | 71.84 | 83.12 | 58.30 | 83.00 | 73.42 |
| $\text{BERT}_{QAT}$ | 90.12 | 80.14 | 84.72 | 67.74 | 85.90 | 82.78 |
| $\text{BERT}_{VT100}$ | 92.32 | 85.86 | 86.22 | 66.14 | 87.50 | 84.92 |
| $\text{BERT}_{VT75}$ | 92.28 | 84.20 | 86.16 | 65.88 | 87.56 | 85.16 |
| $\text{BERT}_{VT50}$ | 92.16 | 85.30 | 86.12 | 60.50 | 87.30 | 84.40 |
| $\text{BERT}_{VT25}$ | 91.46 | 84.82 | 85.24 | 62.54 | 86.78 | 82.34 |

(b) MultiNLI with different binary labels.

Table 3: Model performance averaged across 5 seeds. WGA decreases as model size is reduced in MultiNLI and SCOTUS, but increases instead in CivilComments. This trend is also seen in the binary variants of MultiNLI despite a reduction in task complexity.

| Sparsity | MultiNLI | | CivilComments | | SCOTUS | |
|---|---|---|---|---|---|---|
| | Average | Worst | Average | Worst | Average | Worst |
| 0% | 82.58 | 64.06 | 92.96 | 52.22 | 82.62 | 50.36 |
| 20% | 81.92 | 66.20 | 92.66 | 57.12 | 82.42 | 44.62 |
| 40% | 81.84 | 67.30 | 92.74 | 58.26 | 82.34 | 50.82 |
| 60% | 81.44 | 63.12 | 92.84 | 55.36 | 81.52 | 50.00 |
| 80% | 74.32 | 40.34 | 92.32 | 53.40 | 74.40 | 0.00 |

(a) MultiNLI, CivilComments, and SCOTUS.

| Sparsity | Y = [0, 1] | | Y = [0, 2] | | Y = [1, 2] | |
|---|---|---|---|---|---|---|
| | Average | Worst | Average | Worst | Average | Worst |
| 0% | 92.54 | 85.24 | 86.74 | 67.80 | 87.70 | 84.52 |
| 20% | 92.54 | 85.32 | 86.48 | 69.48 | 87.02 | 82.34 |
| 40% | 92.38 | 85.28 | 86.38 | 69.16 | 87.16 | 84.06 |
| 60% | 91.90 | 84.52 | 85.76 | 66.70 | 86.76 | 84.50 |
| 80% | 86.00 | 70.38 | 78.52 | 38.72 | 81.78 | 76.28 |

(b) MultiNLI with different binary labels.

Table 4: Average and worst-group accuracies for unstructured pruning. $BERT_{Base}$ is shown with a sparsity of 0%. MultiNLI and SCOTUS generally see a worsening WGA when sparsity increases contrary to the improvements in CivilComments.

| Model | MultiNLI | | CivilComments | | SCOTUS | |
|---|---|---|---|---|---|---|
| | Average | Worst | Average | Worst | Average | Worst |
| $TinyBERT_6$ | 85.26 | 72.74 | 92.80 | 52.72 | 77.58 | 29.78 |
| $TinyBERT_{AHE}$ | 77.78 | 53.96 | 92.56 | 52.88 | 77.28 | 38.70 |
| $TinyBERT_{AH}$ | 77.76 | 54.50 | 92.54 | 53.76 | 76.70 | 38.94 |
| $TinyBERT_{AE}$ | 77.14 | 52.24 | 92.56 | 53.22 | 76.82 | 37.48 |
| $TinyBERT_{HE}$ | 75.68 | 51.66 | 92.54 | 51.70 | 77.76 | 39.26 |
| $TinyBERT_A$ | 77.12 | 51.62 | 92.58 | 53.42 | 77.14 | 34.98 |
| $TinyBERT_H$ | 76.08 | 53.12 | 92.52 | 50.82 | 78.06 | 42.04 |
| $TinyBERT_E$ | 65.90 | 32.22 | 92.00 | 44.16 | 75.58 | 33.30 |

(a) MultiNLI, CivilComments, and SCOTUS.

| Model | Y = [0, 1] | | Y = [0, 2] | | Y = [1, 2] | |
|---|---|---|---|---|---|---|
| | Average | Worst | Average | Worst | Average | Worst |
| $TinyBERT_6$ | 94.24 | 89.06 | 89.44 | 76.50 | 89.40 | 87.62 |
| $TinyBERT_{AHE}$ | 88.82 | 78.90 | 81.88 | 51.26 | 84.58 | 82.82 |
| $TinyBERT_{AH}$ | 88.82 | 78.50 | 81.72 | 53.04 | 84.62 | 82.04 |
| $TinyBERT_{AE}$ | 88.12 | 78.32 | 80.86 | 48.78 | 83.90 | 81.58 |
| $TinyBERT_{HE}$ | 87.48 | 75.88 | 80.60 | 49.74 | 83.36 | 80.56 |
| $TinyBERT_A$ | 87.98 | 78.50 | 80.86 | 49.76 | 83.76 | 81.70 |
| $TinyBERT_H$ | 87.76 | 76.26 | 80.94 | 53.80 | 83.58 | 79.80 |
| $TinyBERT_E$ | 78.68 | 61.10 | 72.84 | 38.74 | 72.22 | 65.80 |

(b) MultiNLI with different binary labels.

Table 5: Ablation of $TinyBERT_6$. The subscripts $A$, $H$, and $E$ represent the attention matrices, hidden states, and embeddings that are transferred as knowledge respectively during distillation. A noticeable performance gap is seen between $TinyBERT_6$ and $TinyBERT_{AHE}$ on MultiNLI and SCOTUS.