# OpenReview forum: "Are Compressed Language Models Less Subgroup Robust?"
_EMNLP/2023/Conference — EMNLP 2023 Main_

### Official Review · Reviewer_GktP · 2023-08-04

**Soundness:** 3

**Excitement:**

2: Mediocre: This paper makes marginal contributions (vs non-contemporaneous work), so I would rather not see it in the conference.

**Paper Topic And Main Contributions:**

The paper explores how different compressions techniques for Transformer-encoder models like BERT and ROBERTA affect the classification robustness for minority classes and subgroups. The authors explore 18 different techniques and models available from HuggingFace. An interesting finding is that compression improves subgroup robustness and smaller models are not always worse.

**Reasons To Accept:**

* The paper is well written and the problem is relevant to the NLP community.
* Surveys a large number of techniques and models for compression, and their effect on subgroup robustness.

**Reasons To Reject:**

* No novel techniques or salutations are proposed. This is a survey paper.
* The paper focuses on BERT-style models. It will be more interesting to explore how recent compressions and quantization affects LLM models like LLAMA.
* The paper uses only two datasets. There are many other standard classification datasets that could have been included.
* The subgroup partitioning is not clearly defined. The details are delegated to the appendix but are still not clear.
* Lacks deeper analysis on why TinyBERT is an outlier with regards to subgroup robustness. What are the specific things that make TinyBERT better, some ablation study is warranted.

**Reproducibility:**

3: Could reproduce the results with some difficulty. The settings of parameters are underspecified or subjectively determined; the training/evaluation data are not widely available.

**Reviewer Confidence:**

3: Pretty sure, but there's a chance I missed something. Although I have a good feel for this area in general, I did not carefully check the paper's details, e.g., the math, experimental design, or novelty.

---

> ### Author Rebuttal · Authors · 2023-08-28
>
> Hi, thank you for the feedback regarding the paper. We address your concerns and questions below.
>
> Reasons To Reject:
>
> **No novel techniques or salutations are proposed. This is a survey paper.**
>
> The objective of our paper is to understand how existing methods affect the subgroup robustness of compressed models. We do not propose a novel compression method, which is out of the scope of this paper. By understanding which methods retain the highest subgroup robustness, we will be better able to develop novel solutions from those methods.
>
> **The paper focuses on BERT-style models. It will be more interesting to explore how recent compressions and quantization affects LLM models like LLAMA.**
>
> The field of subgroup robustness research is focused on classification tasks (Sagawa et al., 2020; Koh et al., 2021; Izmailov et al., 2022). As such, generative models like LLAMA are not best suited for this kind of problems, and we consider them out of the scope of our work. Nonetheless, investigating generative tasks is an interesting direction that we would like to pursue in a future work.
>
> **The paper uses only two datasets. There are many other standard classification datasets that could have been included.**
>
> We address this concern in  the official comment.
>
> **The subgroup partitioning is not clearly defined. The details are delegated to the appendix but are still not clear.**
>
> We detail this aspect in the official comment.
>
> **Lacks deeper analysis on why TinyBERT is an outlier with regards to subgroup robustness. What are the specific things that make TinyBERT better, some ablation study is warranted.**
>
> We agree that TinyBERT requires further ablation study on its knowledge distillation process in order to understand why it displays a high subgroup robustness particularly on MultiNLI. At the time, we hypothesized that the more comprehensive knowledge distillation of TinyBERT involving not just the predictions, but also the hidden states, attention matrices, and embeddings may be responsible for its higher subgroup robustness.

---

### Official Review · Reviewer_RMg6 · 2023-08-04

**Soundness:** 3
**Typos Grammar Style And Presentation Improvements:** 1. Giving a brief definition of a con…

**Excitement:**

3: Ambivalent: It has merits (e.g., it reports state-of-the-art results, the idea is nice), but there are key weaknesses (e.g., it describes incremental work), and it can significantly benefit from another round of revision. However, I won't object to accepting it if my co-reviewers champion it.

**Paper Topic And Main Contributions:**

This work conducts an analysis of various compression methods on the subgroup robustness of LMs.
They test 18 different compression methods on MultiNLI and CivilComments. Empirical studies show that
the worst-group robustness relies on both model size and compression method.

**Questions For The Authors:**

1. Why did you choose MultiNLI and CivilComments? The motivation is not clearly stated.
2. The subgroup definition in each dataset is unclear

**Reasons To Accept:**

1. This work first investigates the correlation between the compression method and subgroup robustness
2. The experiments include a wide range of compression methods, which might be useful for future studies

**Reasons To Reject:**

1. Some key descriptions are missing, e.g., concept and abbreviation definition, the selection of datasets, and how to divide each dataset into subgroup partitions. Therefore, this makes it hard to understand the paper
2.  I'm struggling to find the main conclusion in each experiment, especially for Figure 1 and 2

**Reproducibility:**

4: Could mostly reproduce the results, but there may be some variation because of sample variance or minor variations in their interpretation of the protocol or method.

**Reviewer Confidence:**

3: Pretty sure, but there's a chance I missed something. Although I have a good feel for this area in general, I did not carefully check the paper's details, e.g., the math, experimental design, or novelty.

---

> ### Author Rebuttal · Authors · 2023-08-28
>
> Hi, thank you for the feedback regarding the paper. We address your concerns and questions below.
>
> Reasons To Reject:
>
> **Some key descriptions are missing, e.g., concept and abbreviation definition, the selection of datasets, and how to divide each dataset into subgroup partitions. Therefore, this makes it hard to understand the paper**
>
> Thank you for the observation. Given the limited space of the short paper, we did not include a definition of each compression method. However, we will incorporate these definitions in the revised manuscript version. We detailed the choice of datasets  below and the subgroup partitions in the official comment.
>
> **I'm struggling to find the main conclusion in each experiment, especially for Figure 1 and 2**
>
> Regarding Figure 1, we show how different model sizes and compression methods affect the subgroup robustness of a model. We observe that compression does not always worsen the subgroup robustness particularly on datasets that the baseline model overfits on (i.e. CivilComments). In such cases, a reduction in model size can help achieve a better fit on the minority subgroups.
>
> Regarding Figure 2, we break down the performance into subgroups for MultiNLI and CivilComments with knowledge distillation. We show that model compression does not always maintain overall performance by sacrificing the minority subgroups (Hooker et al., 2021, 2020; Stoychev and Gunes, 2022).
>
> Questions For The Authors:
>
> **Why did you choose MultiNLI and CivilComments? The motivation is not clearly stated.**
>
> As mentioned in the Limitations section, L273-274,  MultiNLI and CivilComments are benchmark datasets used by most subgroup robustness research (Sagawa et al., 2020; Koh et al., 2021; Izmailov et al., 2022). Thank you for the observation, we will make it more clear and also add a statement within the paper to better clarify this aspect, which is actually relevant to frame the contribution of the paper.
>
> **The subgroup definition in each dataset is unclear**
>
> See the official comment.

---

### Official Review · Reviewer_gpj9 · 2023-08-05

**Soundness:** 3

**Excitement:**

3: Ambivalent: It has merits (e.g., it reports state-of-the-art results, the idea is nice), but there are key weaknesses (e.g., it describes incremental work), and it can significantly benefit from another round of revision. However, I won't object to accepting it if my co-reviewers champion it.

**Paper Topic And Main Contributions:**

This paper is an empirical study of how different methods of model compression affect performance on minority subgroups. 18 methods are tested on two datasets, and the results show that worst-group performance depends on both the method and model size.

**Questions For The Authors:**

My questions follow my two reasons to reject above:

1. Do you think CivilComments as a dataset is an outlier in that its subgroup performance actually improved?
2. Why do you think certain compression methods work better than others? Is parameter a potential confound here since different methods yield models of different sizes?

**Reasons To Accept:**

1. Many different flavors of model compression methods are tested. The results are definitely helpful for those shopping for more subgroup-robust techniques.
2. Very clearly written and easy to understand.

**Reasons To Reject:**

1. I would have appreciated more datasets, esp given that lots of compressed models actual perform better than the vanilla one on CivilComments. It would have helped to have more results to know whether it's an outlier or genuinely interesting trends.
2. I find the overall conclusion a bit lackluster, esp given the absence of any substantive analysis on why certain methods perform better than others.

**Reproducibility:**

3: Could reproduce the results with some difficulty. The settings of parameters are underspecified or subjectively determined; the training/evaluation data are not widely available.

**Reviewer Confidence:**

3: Pretty sure, but there's a chance I missed something. Although I have a good feel for this area in general, I did not carefully check the paper's details, e.g., the math, experimental design, or novelty.

**Typos Grammar Style And Presentation Improvements:**

I find the 'size' of the markers in Figure 1 unintuitive, esp when you have many different shapes that make it hard to compare.

---

> ### Author Rebuttal · Authors · 2023-08-28
>
> Hi, thank you for the feedback regarding the paper. We address your concerns and questions below.
>
> Reasons To Reject:
>
> **I would have appreciated more datasets, esp given that lots of compressed models actual perform better than the vanilla one on CivilComments. It would have helped to have more results to know whether it's an outlier or genuinely interesting trends.**
>
> We addressed the following issue in the official comment.
>
> **I find the overall conclusion a bit lackluster, esp given the absence of any substantive analysis on why certain methods perform better than others.**
>
> We addressed the question of why certain methods perform better than others further below.
>
> Questions For The Authors:
>
> **Do you think CivilComments as a dataset is an outlier in that its subgroup performance actually improved?**
>
> We address the following issue in the “Additional Experiments” section of the official comment.
>
> **Why do you think certain compression methods work better than others? Is parameter a potential confound here since different methods yield models of different sizes?**
>
> We noted that methods such as post-training quantization (BERT_DQ and BERT_SQ) which do not possess an additional fine-tuning step after compression (e.g. pruning) or a compression-aware training (BERT_QAT) are naturally less subgroup robust. These methods do not allow the model to recover performance after compression or to prepare for compression by learning compression-robust weights.
>
> Directly comparing different compression methods is difficult as different methods compress different parts of the model. For example, the compression of vocabulary transfer is upper bounded by the size of the embedding matrix. However, plotting different compression methods together provides us with a useful general overview of the effects of compression on subgroup robustness.

---

### Meta-Review · Area_Chair_DS3s · 2023-09-19

**Recommendation:** 3

**Metareview:**

This paper explores how the model compression impacts the models' robustness on minority subgroups. The authors present experiments with 18 models compressed by different methods or settings, on 3 datasets (the last one is added during the discussion period).

As the reviewers mentioned, the paper includes a large list of models including knowledge distillation, post-training quantization, quantization-aware training, structured pruning, and vocabulary transfer. This is comprehensive and helpful for future studies. In terms of datasets, the authors provided further context for the dataset selection during the discussion period. Furthermore, they presented another set of experiments on a new dataset addressing reviewers' feedback.

However, also should be noted that reviewers pointed out a lack of analysis on why certain compression methods/models work better than others such as TinyBert. Although the authors attempted to address this during the discussion period, the paper would still benefit from a deeper analysis.

---

### Decision · Program_Chairs · 2023-10-07

**Decision:**

Accept-Main

**Comment:**

This paper explores how the model compression impacts the models' robustness on minority subgroups. The authors present experiments with 18 models compressed by different methods or settings, on 3 datasets (the last one is added during the discussion period).

As the reviewers mentioned, the paper includes a large list of models including knowledge distillation, post-training quantization, quantization-aware training, structured pruning, and vocabulary transfer. This is comprehensive and helpful for future studies. In terms of datasets, the authors provided further context for the dataset selection during the discussion period. Furthermore, they presented another set of experiments on a new dataset addressing reviewers' feedback.

However, also should be noted that reviewers pointed out a lack of analysis on why certain compression methods/models work better than others such as TinyBert. Although the authors attempted to address this during the discussion period, the paper would still benefit from a deeper analysis.